# What Limits Our Capacity to Process Nested Long-Range Dependencies in Sentence Comprehension?

**DOI:** 10.3390/e22040446

**Published:** 2020-04-16

**Authors:** Yair Lakretz, Stanislas Dehaene, Jean-Rémi King

**Affiliations:** 1Cognitive Neuroimaging Unit, Commissariat à l’Energie Atomique (CEA), Institut National de la Santé et de la Recherche Médicale (INSERM) U992, NeuroSpin Center, 91191 Gif-sur-Yvette, France; stanislas.dehaene@cea.fr; 2Collège de France, 11 Place Marcelin Berthelot, 75005 Paris, France; 3Laboratoire des Systèmes Perceptifs, Département D’études Cognitives, École Normale Supérieure, PSL University, CNRS, 75005 Paris, France; jeanremi.king@gmail.com

**Keywords:** sentence processing, long-range dependencies, double center-embeddings, language model, artificial neural networks

## Abstract

Sentence comprehension requires inferring, from a sequence of words, the structure of syntactic relationships that bind these words into a semantic representation. Our limited ability to build some specific syntactic structures, such as nested center-embedded clauses (e.g., “The dog that the cat that the mouse bit chased ran away”), suggests a striking capacity limitation of sentence processing, and thus offers a window to understand how the human brain processes sentences. Here, we review the main hypotheses proposed in psycholinguistics to explain such capacity limitation. We then introduce an alternative approach, derived from our recent work on artificial neural networks optimized for language modeling, and predict that capacity limitation derives from the emergence of sparse and feature-specific syntactic units. Unlike psycholinguistic theories, our neural network-based framework provides precise capacity-limit predictions without making any a priori assumptions about the form of the grammar or parser. Finally, we discuss how our framework may clarify the mechanistic underpinning of language processing and its limitations in the human brain.

## 1. What Are the Computational Mechanisms that Explain Our Limited Ability to Process Specific Sentences?

Sentence comprehension requires the human brain to combine a sequence of word meanings into a continuously evolving semantic representation. The order in which such word combination should occur is non-trivial, because the syntactic organization of sentences is not sequential: words are often connected to one another across long distances. Consequently, the brain must decide, at each time instant, whether and how to store and combine a newly presented word.

The difficulty of this task varies from case to case. Sentences that contain certain long-range dependencies among their words are known to be more demanding to process compared to sentences with only local dependencies. For example, object-extracted relative clauses, such as:

(1)“The **dog** that the cat chased **ran** away.”

contain a center-embedded clause (’the cat chased’), which creates a long-range dependency between the main subject and verb (’dog’ and ’ran’, respectively—in bold). Such long-range dependencies involve a complex memory and processing management which together reduce reading speed and comprehension.

Without specific training, sentence processing dramatically breaks down beyond one level of nesting of relative clauses [1,2]:

(2)“The dog that the cat that the mouse bit chased ran away.”

Such breakdowns show that humans have a strong capacity limitation in sentence processing. Such capacity limitation directly depends on syntax. For example, a sentence that has a highly related meaning, but uses a different word order, is much easier to parse (Figure 1):

(3)“The mouse bit the cat that chased the dog that ran away.”

In other words, the syntactic structure directly impacts our cognitive resources demands beyond their associated semantic complexity. What are the computational mechanisms that explain this capacity limitation during sentence processing? To address this issue, we review several proposals from the psycholinguistic literature. We then introduce an alternative approach to understand capacity limitation in sentence processing, based on our recent work on artificial neural networks optimized for language modeling.

We begin by describing the phenomena and data related to sentence comprehension and the common tools and measures used to quantify sentence-processing loads (Section 2). Next, we describe several cognitive models of sentence processing in psycholinguistics (Section 3). Finally, we describe a new explanation for capacity limitation, which spontaneously emerges from artificial neural networks optimized for language processing (Section 4). To conclude, we outline remaining challenges and open questions in the field (Section 5).

## 2. Empirical Evidence for Processing Difficulties in Sentence Comprehension

Cognitive loads during sentence processing are traditionally estimated by evaluating processing time of each word in a sentence, using eye-tracking or self-paced paradigms. This assumes the more demanding the integration of a word, the longer it takes subjects to process it. For example, Figure 2 presents reading times of two sentence types collected in self-paced reading paradigms. Reading times vary substantially across these two sentences despite their semantic similarity: in the object-extracted relative clauses (blue), readers process the embedded verb ‘sent’ more slowly than when the same verb is processed in subject-extracted relative clauses (orange) [3].

In addition to eye-tracking and self-paced paradigms, processing load can be investigated through question-based paradigms. For example, subjects may be asked to conjugate a verb or to identify a grammatical error in the sentence “The **keys** to the cabinet **is** on the table”. The amount of errors, and the average reaction times obtained in such number-agreement tasks vary as a function of syntactic constructs and thus offer complementary tools to probe processing loads during sentence comprehension [4,5,6,7,8].

Traditionally, studies have assessed and contrasted cognitive models for sentence processing using a small set of examples. More recently, attempts were made to test models on larger sets of items, taken from natural occurring texts. Models can then be evaluated on both their coverage (i.e., their ability to explain a variety of phenomena in language—syntactic complexity, ambiguity resolution, etc.) and their accuracy in explaining collected experimental measures, such as eye tracking and self-paced paradigms [9,10,11].

## 3. Explanations for Syntactic Capacity Limitation by Cognitive Models

Psycholinguistic theories of sentence processing commonly comprise three main elements: (1) a grammar, (2) a parsing algorithm, and (3) a ‘linking hypothesis’ [11]. First, the grammar specifies the latent structure with which words are combined. For example, dependency grammars allow the construction of directed acyclic graphs where each node is a word, and each edge is a syntactic link. By contrast, constituency grammars allow undirected graphs where each node is a syntactic construct (a verb phrase, noun phrase, etc.). Second, the parsing algorithm describes the computational steps of sentence processing. Such algorithm thus generates the syntactic structure from a sequence of words. Finally, the linking hypothesis postulates specific resource limitations during sentence processing.

In what follows, we describe three main families of cognitive models for sentence processing—expectation-based, memory-based and symbolic neural architectures theories—and how capacity limitation is explained by each one of them.

### 3.1. Memory-Based Theories

#### 3.1.1. Dependency Locality Theory

The Dependency Locality Theory (DLT, [12]) was proposed to account for *locality* effects. Locality effects refers to subjects’ tendency to combine adjacent word pairs, as opposed to distant ones. For example, readers report semantic anomalies when reading “Mary said that the kids will swim yesterday” since the last item ‘yesterday’ would, according to the locality effect, be attached to the most recent phrase (“the kids will swim”) rather than to the main clause (“Mary said”) [3]. Similarly, in sentences with three verb phrases, for example, “The judge noted that the guard confirmed that the prisoner confessed the crime last week”, readers find increasing difficult to link the modifier ‘last week’ with the last, middle and first VP [13], in that order. These locality effects motivate the hypothesis that longer dependencies are harder to process compared to local ones.

To model locality effects, DLT assumes dependency analysis and postulates the existence of *energy units* and two processing costs: *storage* and *integration* costs. Energy units are required for sentence processing. The number of energy units required for the processing of a given word is determined based on the storage and integration costs, which in turn, depend on syntactic complexity, previous items to be stored, and so forth. The processing time of each word can then be inferred according to: EnergyUnits=MemoryUnits∗TimeUnits. For example, when the storage and integration costs are high (i.e., a high number of energy units is required), more time units will be required for the processing of a word, assuming that at each time point there’s a limited number of available memory-units.

In complex sentences, processing costs may be exceptionally high, leading to excessive processing times at specific words—Box 1 illustrates the integration costs predicted by DLT in sentences with double center embeddings (2).

Box 1The Dependency Locality Theory (DLT).

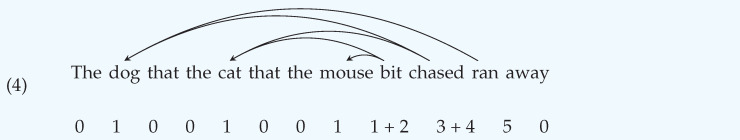

**Integration Costs in the Dependency Locality Theory (DLT)** we illustrate integration costs in a doubly center-embedded sentence (4) as described by the DLT [3,14]. First, *discourse referents* in DLT are either discourse objects (e.g., nouns as ‘dog’) or tensed verbs (e.g., ‘ran’). The integration cost at each time point is defined to be proportional to the number of new discourse referents that are introduced between the endpoints of an integration. For example, the integration costs at ‘dog’, ‘cat’, ‘mouse’ are all equal to 1, since they are local, having only a single discourse referent between the endpoints of the dependency. Next, the integration at ‘bit’ involves two dependencies (illustrated by the two dependency arcs in (4)): one with respect to the subject (‘mouse’), and the other with respect to the object (‘cat’). The dependency to the subject traverses a single discourse referent (an event of biting), and thus its integration cost is equal to 1. The dependency to the object traverses two discourse referents (an event of biting and ‘mouse’), and thus its cost is equal to 2. Consequently, the total integration cost at ‘bit’ is equal to 3. Similarly, at ‘chased’, the corresponding costs are 3 and 4. Finally, the integration at ‘ran’ is only for the dependency with the preceding subject (‘dog’), which traverses five discourse referents. Taken together, this predicts a processing profile that has its highest peak at the intermediate verb of the sentence. Processing breakdown is predicted at this point, surpassing memory capacity, given limited processing resources.

#### 3.1.2. ACT-R Based

A second line of research couches sentence processing within the more general framework of working memory—the Adaptive Control of Thought–Rational (ACT-R) [15]. In accordance with ACT-R, Lewis and Vasishth [16] argues that during incremental sentence processing, the syntactic structure is stored incrementally in a content-addressable memory. Content-addressable memory means that information is retrieved from working memory based on its content rather than, for example, its location. At each time step, the transient syntactic structure is thus assumed to be encoded in working memory across several memory items, or ‘chunks’ (Box 2).

During sentence processing, each new word triggers a memory retrieval. At the end of the retrieval, the new word is integrated into one of the memory chunks that encodes the current syntactic structure. Since memory is assumed to be content-addressable, retrieval is done based on the similarity between cues and stored chunks. Such retrieval can thus lead to *interference* with other, unrelated but nonetheless similar, competing chunks. Tight competition leads to slower processing, or to an erroneous integration of the new word in memory, which itself leads to to an incorrect parse.

There are several key differences between DLT and the ACT-R based model, with respect to both their assumptions and predictions. First, while DLT assumes a dependency grammar and parsing the ACT-R based model assumes phrase-structure grammar and left-corner parsing [16]. Second, the ACT-R based model suggests more refined predictions in some cases. For example, unlike DLT, it predicts reduced processing difficulties in relative clauses, in which the two subjects are of different types—for example, when one subject is animate (e.g., ‘boy’) and the other is inanimate (‘table’), or when one of the subjects is a pronoun. While DLT predicts processing difficulties solely based on distance, the ACT-R based models predict that difficulties vary with inter-subjects similarity [16], as was found experimentally [8]. Finally, in contrast to DLT, capacity limitation in the ACT-R based model primarily arises due to retrieval interference, rather than resource depletion.

Box 2The ACT-R Based Model.**Encoding of a Syntactic Structure in the ACT-R based Model** In the ACT-R based model, the transient syntactic structure is represented in the model across memory ‘chunks’ in declarative memory [16]. Figure 3 illustrates a chunk representation of a syntactic X-bar structure [17], in which nodes of the tree are mapped onto memory chunks having features that correspond to X-bar positions (head, comp, specifier) and syntactic features (e.g., grammatical number).Each memory chunk is assigned an activity that is enhanced if the chunk was retrieved from memory, otherwise, it decays over time. The access to a given memory chunk during memory retrieval depends on its transient activity.The syntactic structure is not maintained in memory as a unified entity, but it is rather distributed across chunks, each with a different activity. Also, word order is not explicitly represented in the model, but rather indirectly encoded through activity decay. This entails complex dynamics. During incremental processing, each word cues a memory retrieval, after which the new word will be integrated into a memory chunk. During this process, similar chunks interfere, which might lead to processing slowdowns and errors.

**Figure 3 entropy-22-00446-f003:**
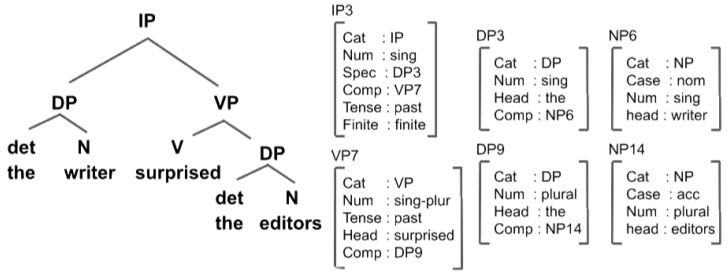
A chunk representation of a syntactic structure in Adaptive Control of Thought–Rational (ACT-R) (adapted from [16]).

### 3.2. Expectation-Based Theories

Expectation-based theories propose that cognitive loads during sentence processing depend on subjects’ expectations before the processing of a given word—that is, unexpected words are more difficult to process than expected ones. Expectations, it is argued, depend on the statistics of natural language, which are captured in a probabilistic grammar formed during language acquisition. Common syntactic structures are more predictable by the grammar, which in turn, makes them more effortless to process. In contrast, relatively rare structures are more difficult to process and thus increase the risk of processing breakdowns.

To quantify expectation, Hale [18] suggested the Shannon information content, or ‘surprisal’, as a measure for processing difficulty: $surprisal\left(w_{i+1}\right)=-log(p\left(w_{i+1}|w_{1}\ldots w_{i}\right)$, where $w_{i}$ are words in the sentence (for illustration purpose, we assume that the sentence is presented without a context). The more probable a word is given a context, the simpler it is to be processed. To estimate surprisal, one must assume a probabilistic grammar, such as a probabilistic context-free grammar (pCFG; [18,19,20]), from which surprisal for each word can be derived (Box 3).

In expectation-based theories, the origin of processing difficulty is rooted in the probabilities of the pCFG. Conesequently, probability mass can be considered as a limiting processing resource, which in turn, entails capacity limitation. In particular, since the posterior distribution $p(T|w_{1},\ldots ,w_{i})$ (Box 3), which is defined over all possible transient syntactic structures given a context, should sum to one, frequent structures in the language can be thought of as ‘consuming’ most of the processing resources at each point in time. This thus leads to high surprisal and an increased processing difficulty when subjects encounter rare syntactic structures.

In the case of center-embedding, the production probability of object-relative clauses in a pCFG is considerably low compared to, for example, subject-relative clauses. Consequently, center-embeddings lead to high surprisal, and, a fortiori, the nesting of two center-embeddings (e.g., (2)) leads to considerable surprisal.

### 3.3. Symbolic Neural Architectures

Smolensky [21] suggested that syntactic structures can be represented with high dimensional vectors. His theory formally shows that the generation of such structures can be computed with tensor products between vectors coding for words, position and role. The major limitation to such tensor product is that the number of dimensions necessary to store a complex structures exponentially grows with syntactic complexity. Smolensky’s theory therefore suggests that capacity limitation in multi-level nesting would result from the limited number of neurons available to process sentences.

Subsequent proposals, such as Vector Symbolic Architectures [22], Binary Spatter-Coding [23], Holographic Reduced Representation [24], proposed hand-crafted solutions to efficiently approximate tensor products with a limited number of neurons. However, because these models are not committed to a specific grammar or parsing algorithm, they actually fail to make specific predictions about why a specific syntactic constructs leads to behavioral deficits in humans.

Box 3Expectation-Based Models.
**Probabilistic Context-Free Grammars (pCFGs)**
A context-free grammar (CFG) is a formal model for how sentence strings are generated in a given language. A CFG comprises of a set of *production rules*, defined over three types of symbols: *terminal* symbols Σ, *non-terminal* symbols *N*, and a start symbol *S*. A *derivation* of a string, is a sequence of rule applications that transforms *S* into a sentence string, composed of terminal symbols only. A derivation can be represented with a *parse tree*, which we denote by *T*.A probabilistic context-free grammar (pCFG) is a CFG complemented with probabilities assigned to all production rules. Below is an illustration of a simple pCFG with Σ= {‘away’, ‘cat’, ‘chased’, ‘dog’, ‘ran’, ‘the’} , and N={S,NP,VP,N,V,Det,P}:

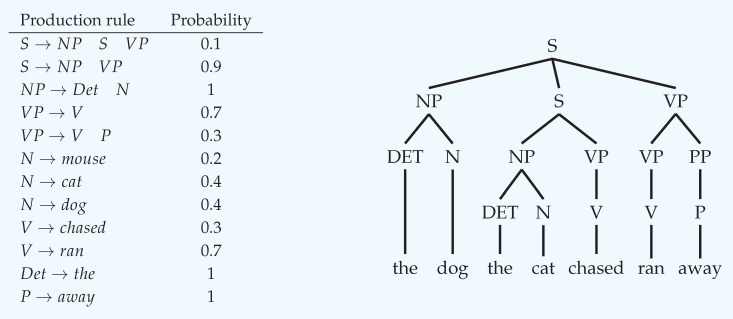

Note that for a given non-terminal on the left-hand side, all probabilities on the right column sum to 1. As an example, the following sentence can be sampled from the pCFG as follows: S→NPSVP→NPNPVPVP→DetNDetNVVP→
thedogthecatchasedranaway
The corresponding probability of this derivation p(T) can be calculated by multiplying the probabilities (right column of the pCFG) of all substitutions performed above. Finally, the generation probability of the sentence, is the sum of the probabilities of all admissible derivations.**Estimation of Surprisal from a pCFG** We show how ‘surprisal’ of a word can be calculated from a given pCFG: let *T* denote a possible transient syntactic tree up to word $w_{i}$, the conditional probability of a new word given its context would be: $p\left(w_{i+1}|w_{1},\ldots ,w_{i}\right)=\sum_{T}p\left(w_{i+1}|T,w_{1},\ldots ,w_{i}\right)p\left(T|w_{1},\ldots ,w_{i}\right)\propto \sum_{T}p\left(w_{i+1}|T,w_{1},\ldots ,w_{i}\right)p\left(w_{1},\ldots ,w_{i}|T\right)p\left(T\right)=\sum_{T}p\left(w_{i+1}|T,w_{1},\ldots ,w_{i}\right)p\left(T\right)$, where the first transition is after marginalizing over all possible transient syntactic trees, the second follows Bayes rule, and the third uses the fact that given a parse tree *T* there’s only a single admissible possibility to generate a sequence of words $w_{1},\ldots ,w_{i}$ (i.e., $p(w_{1},\ldots ,w_{i}|T)=1$).For example, at the third word in “The dog the cat chased ran away”, ‘the’, only a single derivation is admissible, in which the relatively rare center embedding rule (S→NPSVP;p=0.1) appears. There’s no other derivation that can account for the appearance of ’the’ at this point, which does not include this rule. Given that this is the only admissible derivation, $p(w_{i+1}|T,w_{1},\ldots ,w_{i})$ will be zero for all other possible trees. Now, since the only possible derivation includes the application of the center-embedding rule, p(T) would be relatively small since p=0.1 is one of its factors. Consequently, p(the|T,the,dog) would be small, and the surprisal high.To conclude, surprisal, and therefore processing difficulty, primarily arise in expectation-based theories from the production probabilities in the pCFG. This is therefore in stark contrast to memory-based theories, in which processing difficulty arises from resource depletion or interference.

## 4. Understanding Capacity Limitation in Light of Neural Language Models

We suggest here an alternative theoretical framework—taking the recent advances in the deep learning of natural language processing seriously, and consider the resulting artificial neural language models (NLMs) as plausible models of sentence processing. Unlike traditional theories, NLMs make no a priori assumptions about the form of the grammar or the parser, and provides detailed predictions about neural dynamics compared to previous theories, which can be directly assessed with brain data. NLMs describe syntactic and linguistic processing at a much lower description level compared to common models in psycholinguistics. They could therefore be interpreted as a plausible implementation of the cognitive operations our brain generates during sentence comprehension [25]. While neural language models have a very large amount of (hyper-) parameters, they actually make fewer assumptions about sentence processing. Neural language models are shaped and optimized to predict the next word in a training corpus. Consequently, these models make no assumptions specially tailored to account for capacity limitation (Box 4).

### 4.1. NLMs Predict that Capacity Limitation Results from a Functional Specification of Dedicated Syntax and Number Units

In a recent work [26], we studied the processing of long-range dependencies in an NLM and found that a sparse neural circuit emerged in the neural language model during training, which was shown to carry grammatical-number agreements across long-range dependencies in various sentences. In this section, we briefly describe the main findings therein, and describe how a capacity limitation can further be derived.

#### 4.1.1. A Sparse Neural Circuit for Long-Range Number Agreement

The neural circuit for grammatical-number agreement was found to be composed of three types of units with different functionality each:**Syntax units**: units in the network whose activity is predictive of transient syntactic properties of the sentence. A particular set of syntax units was found to be predictive of the transient depth of syntactic tree, which is an index of syntactic complexity [27]. The activity of one of the syntax units was found to follow the structure of the main subject-verb dependency in various sentences (green curve in Figure 4. See also Figure 3 in Ref. [26]). Specifically, the activity of this syntax unit is positive throughout the main subject-verb dependency and changes sign only at the main verb. As discussed further below, this activity profile allows the carrying of the grammatical number of the main subject across long distances.**Long-Range number units**: units that can encode grammatical number (singular or plural) for long-range dependencies. Out of total of 1300 units in the neural network, only two were identified as long-range number units—one unit for singular and the other for plural number. Long-range number units were shown to encode the grammatical number of the main subject (‘keys’ in “The keys to the cabinet are..”) and robustly store it up to the main verb (‘are’) across possible attractors (‘cabinet’). Figure 4A illustrates the activity profile of the singular (red) and plural (blue) long-range units during the processing of a subject-extracted relative clause whose main subject is singular. Note that while the activity of the plural unit is silent (i.e., around zero), the singular unit is active throughout the subject-verb dependency, and beyond the attractor. Figure 4B illustrates the ‘mirror’ case, in which the main subject of the sentence is plural. Importantly, ablation of any of the long-range number units was shown to bring the performance of the network on number-agreement tasks to chance level on the corresponding task. In sum, the long-range units carry out the long-range number agreement task in the network.**Short-Range number units**: units that encode the grammatical number of the last encountered noun. In contrast to long-range units, short-range number units can carry grammatical number only across dependencies that do not contain intervening nouns with opposite numbers. Short-range number units were identified using decoding methods [28].

Importantly, a network connectivity analysis revealed that the syntax unit had exceptionally strong efferent connections to both long-range units, compared to all other units in the network, which was shown to be crucial for the writing and erasing of grammatical number information in both long-range units. Taken together, the three units (singular, plural and syntax unit) were found to form a neural circuit dedicated to the processing of long-range number-agreement dependencies: whenever a long-range dependency occurs, the syntax unit traces it by its activity and conveys this information to the long-range number units, which in turn store and carry the grammatical number of the subject up to the main verb.

We highlight the fact that long-range dependencies were thus found to be processed by a very small number of units in the network, namely, three. The *sparsity* of the mechanism for long-range number agreement entails a capacity limitation to process multiple dependencies as we next show.

Box 4Neural Language Models.
**Neural Language Models (NLMs)**
A language model is a probabilistic model that assigns a probability to any sequence of words from a given alphabet. *Neural* language models are language models that are implemented with artificial neural networks (ANNs), typically, recurrent or attention-based ANNs [29,30,31]. NLMs are commonly trained on a word-prediction task—predicting word probabilities given a context. The models are typically trained on a large non-annotated corpora. To perform the word-prediction task, the model learns the statistical regularities in natural language. Since the corpus is non-annotated, complex linguistic constructs such as clauses, subject-verb agreement must be implicitly inferred by the network. Ultimately, this linguistics knowledge is stored in its ‘synaptic’ connections. This knowledge can thus be probed at test time to infer what the network has learned and how it processes sentences.**Probing Syntactic Processing in NLMs** Research in natural language processing has made remarkable progress over the past decade. In particular, NLMs have repeatedly broken records on language modeling and back-translation (e.g., evaluating the similarity between an English sentence, and its translation in German and back into English). However, the remarkable performance of NLMs comes at a price: their very large amount of parameters currently limit our ability to understand *how* they process sentences.Two complementary approaches address this interpretability challenge. First, one can study the behavior of a neural network [32,33,34]. Specifically, sets of carefully designed sentences are presented to a trained neural network, and one can subsequently measure error rates to infer processing load, similarly to experiments with human subjects.For example, in a seminal work, Linzen et al. [32] explored long-range agreement in NLMs. Agreement is defined as the copy of features of one word onto another, such as grammatical number or gender, and is considered as one of the best indexes of the syntax-internal dynamics [6]. In subject-verb agreement:(5)The **keys** to the cabinet
**are** on...The subject ’keys’ and the verb ’are’ agree on their grammatical number (plural). Linzen et al. [32] presented to the model a sequence containing all words up to the verb (exclusive), also known as the ‘preamble’, and evaluated the model by testing whether it assigns a higher probability to the correct form of the verb (‘are’) than to the incorrect one (‘is’). Error rates were then collected across a large number of such sentences. This approach has identified similarities between humans and NLMs. For example, it is known that in humans, the presence of an intervening noun with an opposite grammatical number (‘cabinet’ in (5)) elicits more agreement errors, and the more attractors occur between the subject and the verb the more agreement errors people make [4]. Such patterns were found also in NLMs [32]. In contrast, differences in sentence processing between humans and neural models were also been reported. For example, while in humans, the occurrence of an attractor within a prepositional phrase was found to elicit more agreement errors compared to its presence in a relative clause [4], recurrent neural network language models were found to show the opposite effect [35]. Similarly, recurrent neural models were found in Reference [33] to produce different errors rates than humans on a variety of syntactic structures (but see, References [26,36]).Analyzing the behavior of the network, however, leaves the underlying computational mechanism unexplained. A second approach thus consists in studying the activation patterns in the network and link them to meaningful computational constructs [37]. For example, using linear classifiers at each time point during sentence processing, information represented by various units can be decoded, and thus provide evidence about their processing function. Using such ‘Diagnostic Classifiers’ [38], Giulianelli et al. [39] explored whether grammatical-number information can be decoded from transient state of a neural network. This approach proved beneficial: in particular, it revealed that the representation of grammatical number, as in (5), is mostly stored by the highest layer of the network, and that it can be robustly maintained in network activity.

#### 4.1.2. Processing of Nested Long-Range Dependencies and Capacity Limitation in NLMs

While the described neural circuit is dedicated to long-range number agreements, the short-range units can handle short-range number agreements, as long as there are no intervening nouns with opposite grammatical number between the subject and verb. The network can therefore successfully process sentences in which a short-range dependency is nested inside a long-range one. For example, in object-extracted relative clauses (1) there are two nested number agreements to be processed: (a) an outer agreement between the main subject and verb, and (b) an inner agreement between the embedded subject and verb (‘cat’ and ‘chased’). In the neural network, the outer dependency can be processed by the long-range number-agreement mechanism (Figure 4), and the inner agreement can be carried out by the short-range number units. Indeed, although model performance in the case of object-extracted relative clauses is relatively low compared to other syntactic structures, similar to humans [40], the network achieves above chance-level performance on both verbs [41].

However, the sparsity of the long-range number-agreement mechanism begs the following question—how could the network process sentences with two nested *long-range* dependencies? Once the syntax unit encodes and follows the outer dependency, and the long-range number units encode the outer grammatical number of the main dependency, the long-range number-agreement mechanism is occupied. There are therefore no more units to represent the nested long-range dependency.

Let us turn to the example of doubly-embedded relative clause. In (2) there are three dependencies, two of which are long-range ones: (1) the outermost between ‘dog’ and ‘ran’, and (2) the middle, nested, long-range dependency between ‘cat’ and ‘chased’. The third dependency, the innermost, is a short-range one: between ‘mouse’ and ‘bit’. While the innermost short-range dependency can be processed by the short-range units and the outermost one by the long-range mechanism, the middle long-range dependency cannot be correctly processed given that the long-range mechanism is already occupied.

The sparsity of the long-range agreement mechanism thus provides a mechanistic explanation for capacity limitation in syntactic processing. During training, the network specializes a small number of units for the processing of long-range dependencies. Since nested long-range dependencies are rare in the training data, a sparse mechanism can suffice to handle long-range dependencies in the majority of the cases. Increasing the number of units in the network therefore does not remove the sparsity of the long-range mechanism. The capacity limitation of the network arises from a limited resource of specialized syntax and long-range number units, and not from a limited resource of ‘general’ units—Networks of different sizes were all found to develop a similar sparse mechanism [42] (in this recent work, we have also found that the long-range unit may possibly encode multiple values through a "fractal" mechanism: in the case of deep nested long-range dependencies, we found that the grammatical noun of all subjects can be encoded in various scales of the long-range unit activity. However, such a mechanism is also inherently limited: the long-distance word is encoded by a much smaller difference in neural activity than the more recent word, and therefore is more sensitive to noise. In the end, therefore, this fractal mechanism also predicts an increasing difficulty as the number of embeddings increases). Table 1 summarizes the main elements and explanations for capacity limitation for all described theories (Section 3) and for NLMs.

#### 4.1.3. Capacity Limitation also Emerges in the Case of Deeper Nesting

In a recent study [42], we further studied the capabilities of NLMs to learn a simple artificial grammar for recursive patterns with long-range dependencies. One motivation to study syntax processing with artificial grammars is that it avoids difficulties that arise when studying natural data, such as correlations between syntactic and semantic information. Furthermore, artificial grammars allow to carefully control for various parameters, such as occurrence statistics of certain structures in the data.

In this study, using probabilistic Context-Free Grammars (pCFGs), we generated artificial languages having center-embeddings. The languages contained subject-verb dependencies with number agreement. Importantly, the languages differed by their statistics, which were controlled by two parameters $p_{1}$ and $p_{2}$: $p_{1}$ determined the generation probability of an additional level of center-embedding nesting—a larger value of $p_{1}$ generated languages with deeper nestings; $p_{2}$ determined the generation probability of an adverb-like token, which did not carry grammatical number—a larger value of $p_{2}$ generated languages with longer dependencies.

Using this framework, we then studied whether a NLM trained on a language with given statistics (e.g., with ’shallow’ nestings) can generalize to a different one that has more challenging structures—deeper nesting or longer dependencies. By independently manipulating $p_{1}$ and $p_{2}$ we could decouple generalization patterns to deeper structures from those to longer dependencies. We further studied whether NLMs develop a sparse mechanism, as was found in English, also when trained on languages with deeper structures that those occur in natural language.

Languages were generated from a large range of $p_{1}$ and $p_{2}$ values, and various types of NLMs were trained on each one of them. Several of the NLMs were models with a structural bias towards hierarchical processing (e.g., stack-LSTMs [43]), hypothesizing that structurally-biased models would better capture deep nested dependencies. Results showed that several models generalized to unseen datasets having longer dependencies than those seen during training. However, crucially, capacity limitation to process nested dependencies was consistently observed in all explored NLMs. None of the models, nor the structurally-biased ones, truly captured the underlying recursive regularity. All models showed good generalization to unseen sentences having similar nesting depth, but poor generalization to unseen sentences having deeper nested structures compared to those seen during training. Furthermore, a detailed analysis of one of the NLMs revealed a sparse mechanism to encode multiple grammatical numbers in a single unit. This provided further evidence for the emergence of a sparse mechanisms for long-range dependencies in LSTM language models.

#### 4.1.4. Varying Processing Difficulties of Sentences with Two Long-Range Dependencies

Behavioral experiments with humans showed that some sentences with two long-range dependencies can be easier to process compared to doubly center-embedded sentences. One such case is an embedding of a relative clause (RC) within a sentential complement (SC), for example: “The fact that the employee who the manager hired stole office supplies worried the executive (SC/RC)”, which was shown to be easier to process than the reversed embedding (RC/SC) and doubly center-embedded sentences [12]. Another example is doubly center-embedded sentences as (2) in which the innermost subject is replaced by a personal pronoun, which was found to facilitate processing.

In contrast, the above explanation for capacity limitation in NLMs relies on the observation that these models cannot simultaneously process two long-range dependencies. Consequently, NLMs do not predict a variation of processing difficulty as a function of such properties. However, varying processing difficulties can naturally occur in NLMs: for example, in the case of object-extracted relative clauses, we found that inter-subject similarity affects processing. Specifically, we compared the error rate of the NLM explored in Reference [26] on object-extracted relative clauses with two nouns (“The boy that the mother likes is smiling”) and on object-extracted relative clauses with one subject noun and one personal pronoun (“The boy that she likes is smiling”), and found that the error rate was higher in the former case by 22%. Such reduced processing difficulty in the case of smaller inter-subject was pointed out as one of the advantages of the ACT-R based model compared to DLT [16], and is consistent with similar findings in humans, e.g., [8]. Nonetheless, further research is needed to explore varying processing difficulties in NLMs in the case of two long-range dependencies.

Finally, we note that NLMs partially relates to the classic Augmented Transition Network (ATN) [44]. ATN originally made similar predictions regarding the processing of two long-range dependencies that are active at once and thus faced similar challenges. Specifically, in ATN, a long-range dependency was handled by using a pointer to a “HOLD” register. Once a dependent is encountered, a pointer to “HOLD” is given, without regard to when the dependency would end up, which resembles the encoding dynamics of the syntax unit. Then, once the verb is presented, the pointer is retrieved from HOLD, which is similar to the drop in activity of the syntax unit once the verb is encountered. Assuming a limited memory, it was unclear how to generalize the HOLD register to handle multiple long-range dependencies, which is similar to the case in NLMs. However, unlike the ATN, processing difficulties on various constructions in NLMs may depend on various factors such as their statistics in the training corpus. Further research is needed to explore such effects.

### 4.2. Neural Language Models Make Precise Predictions on How Humans Process Sentences

Several predictions in humans can be derived from the neural language model, both with respect to behavior and cortical processing:

Behavioral predictions:

**Embedded dependencies are more error prone:** the main, ‘outer’, dependency is processed by the long-range mechanism, which was shown to be robust, protecting the carried grammatical number from intervening information. Embedded dependencies, however, are more susceptible to interference and processing failures. Indeed, we found that both humans and neural language models make more number-agreement errors on the embedded verb of center-embedded clauses compared to the main one [41]. In this study, humans and neural networks were tested on both simple object-extracted relative clauses (1) and object-extracted relative clauses in which the inner dependency is also a long-range one (“The dog that the **cats** near the car **chase** runs away”). Both humans and neural networks make more agreement errors on the inner verb (‘chase’) compared to the outer one (‘runs’).**Agreement-error patterns show primacy and recency effects:** In doubly center-embedded sentences, while the main agreement can be robustly processed by the long-range mechanism, and the innermost by short-range units, the middle long-range dependency is predicted to be most error prone. In a recent study [42], we tested agreement errors of a variety of neural language models trained on deep nested long-range dependencies generated from an artificial grammar. Neural language models were indeed found to make more number-agreement errors on middle dependencies compared to the outer-most and innermost ones, showing a ‘recency and primacy effects’ in the error patterns. The model therefore predicts similar agreement-error patterns in human behavior. We note that this error pattern is consistent with ’structural forgetting’, a phenomenon reported in humans, e.g., [45], in which English speakers tend to judge ungrammatical sentences with doubly center embedding and a missing verb as grammatically correct (e.g., “The patient who the nurse who the clinic had hired met Jack”). Importantly, structural forgetting occurs with sentences in which the middle verb is omitted.

Neural predictions:

**Long- and short-range units reside in different cortical regions:** the two types of units that emerged in the models during training suggest that a similar division may be observed in cortical processing. In particular, Dehaene et al. [46] have suggested a hierarchy of cortical processing in language, from low-level transition regularities to high-level structured patterns across the cortex. In accordance with this view, long-range units are predicted to be found at higher-level cortical regions, and short-range units in lower-level ones, or both (in NLMs trained on natural-language data, long-range number units tend to emerge in the highest layer of the network [26,41], whereas short-range units can be found across several layers). Since long-range units are predicted to be sparse they might be found in highly localized regions of the brain (note that the small number of syntax and number units that emerged in the network is not a realistic estimation of the number of corresponding neurons in the brain, without taking in consideration several corrections. First, the total number of units in the NLM is several orders of magnitude smaller than that in the brain. Second, the NLM is commonly considered as a ‘rate model’. Consequently, the activity of a single unit in the model in response to a feature would map to a large number of spiking neurons in the brain, all responsive to the same feature. Taken together, a single unit in the NLM could therefore correspond to possibly more than $10^{6}$ neurons in the brain. The ‘sparsity’ of the mechanism should therefore not be construed as an extreme localist, ‘grandmother cell’, e.g., [47], view).**Specific syntactic cortical regions show persistent activity throughout long-range dependencies**: the activity pattern of the syntax unit during sentence processing suggests similar dynamics in cortical regions related to syntactic processing. Specifically, activity of certain syntactic regions is predicted to persist throughout a long-range dependency, in order to gate feature-information storage (grammatical number, gender, etc.) in other regions.**Specific syntactic cortical regions project onto long- but not short-range units:** while the activity of long-range units was found to follow the structure of the long-range dependencies, as conveyed by the syntax unit, the activity of the short-range units is not structure sensitive. Neural activity related to syntactic processing of long-range dependencies, presumably in syntactic brain regions, is predicted to drive neural activity related to long-range encoding of grammatical number, but not that related to short-range encoding.

In sum, neural language models are shown to provide precise and testable predictions about both human behavior and its underlying cortical mechanisms, therefore serving as appealing models for both cognitive and mechanistic aspects of human linguistic performance. However, neural language models remain nevertheless ‘toy models’ of the human brain. Moreover, so far, we presented evidence from grammatical agreement only. Although grammatical agreement is considered to be an excellent probe into syntactic processing, e.g., [4,6,7], it can only provide a limited window into language processing in NLMs. In the next section, we further discuss several limitations of NLMs and remaining challenges.

## 5. Remaining Challenges

Artificial neural networks may, as a first approximation, be useful guides to understand how syntactic information is encoded in the brain of human adults. Nonetheless, these toy-models remain fraught with important limitations.
**Language acquisition:** artificial neural networks fail to capture the speed and ease of language acquisition in human children. Children are known to acquire language from a relatively small number of stimuli compared to the complexity of the task of inferring underlying regularities in natural data [48]. In contrast, current state-of-the-art neural language models require large amount of training data, which needed to be presented to the model several times. Finding the structural and learning biases required to reduce such data thirst down to a profile similar to humans, remains one of the major challenges in the field.**Compositionality:** neural language models fail to achieve systematic compositionality [49]. Human language is characterized by systematic compositionality—people can understand and produce a potentially infinite number of novel combinations from a finite set of known elements [50,51]. For example, once an English speaker learns a new verb, for example, “dax”, he can immediately understand and produce sentences like “dax again” or "dax twice”. This algebraic capacity, famously argued to be a property of symbolic-based systems but not of neural networks [52,53], remains a challenge also for modern neural language models. Consequently, it is thus expected that current neural language model will present strong limitations in their ability to account for brain responses to new words.**Fit to brain data:** neural language models fail to achieve good fit to neuroimaging and electrophysiological data (e.g., fMRI, MEG, intracranial). Both language models and neural language models have been used in recent years to model brain data acquired from subjects engaged in naturalistic tasks, such as active listening to stories, e.g., [54,55,56,57]. Typically, activations of language models during sentence processing are extracted and used as predictors in linear regression models, which are used to fit brain data. However, despite preliminary success, current models still explain a relatively small variance in brain data. This suggests that additional—and possibly critical—processes to those of artifical neural networks, are engaged in the human brain when it processes sentences.**Biological plausibility:** although inspired by findings in neuroscience, various aspects of the dynamics and learning in common neural language models cannot be directly mapped onto biologically-plausible mechanisms in the brain. For example, while learning and plasticity in the brain are known to occur locally [58,59], the back-propagation algorithm used to train neural networks [60] relies on error propagation across distant parts of the network. On the other hand, neural-network models that are more faithful to brain dynamics are still hard to tame, e.g., [61,62] and can simulate only limited aspects of linguistic phenomena. Constructing a neural language model that is both biological-plausible and achieves high performance on linguistic tasks remains another challenge in the field.

## 6. Summary and Conclusions

What is the computational explanation for our limited ability to process sentences with nested dependencies? We described several answers for this question from the psycholinguistic literature. In a nutshell, resource-based theories, such as the Dependency Locality Theory [12], postulate that sentence processing depends on a limited computational resource, in the form of, for example, energy units. Capacity limitation therefore arises due to resource depletion, which occurs during the processing of complex sentences. Retrieval-based theories [16] suggest that during sentence processing, each word triggers a memory retrieval process, which terminates in its integration into an incrementally constructed parse tree in memory. High interference among words might render this retrieval process noisy, leading to parse slow-downs or errors. Expectation-based theories postulate that sentence processing is directly shaped by the statistics of natural language to which subjects are exposed. Specifically, the likelihood of each linguistic structure directly modulates subjects’ expectations—unlikely syntactic structures are unexpected and might therefore lead to processing slow downs or breakdowns. Symbolic neural architectures suggest that syntactic structures are represented in high-dimensional vector spaces, which correspond to neural activity. Their finite dimensionality entails a capacity limitation with respect to the complexity of syntactic structure that can be represented without interference.

Finally, we suggested a new answer to this question, based on our recent experiments with neural language models. During training, a sparse mechanism for processing long-range dependencies emerged in the network. The sparsity of this mechanism limits the number of nested, or cross, dependencies that the model can process. Unlike psycholinguistic theories, the neural language model is assumption free with respect to specific resource limitation, grammar or parsing algorithm. The model makes predictions both about human behavior and the underlying cortical processing, which are testable with neuroimaging and electropyhsiological techniques. Despite remaining challenges and yet preliminary empirical evidence, we argue that contemporary NLMs are appealing models for both cognitive and mechanistic aspects of human linguistic performance. They provide means to bring together theories from psycholinguistics, neuroscience and natural language processing, and the integration of knowledge and tools from these different fields. Future work, we argue, could benefit from such an integrative approach in the study of sentence processing in humans.

## Figures and Tables

**Figure 1 entropy-22-00446-f001:**
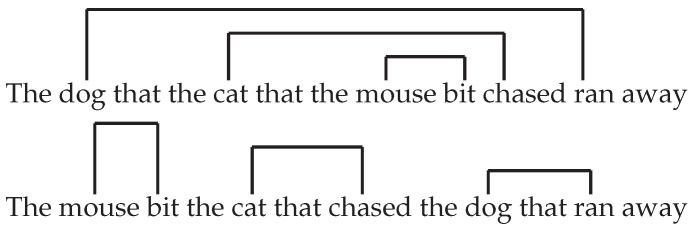
Two sentences that carry the same meaning, with (**top**) and without (**bottom**) double center-embedding.

**Figure 2 entropy-22-00446-f002:**
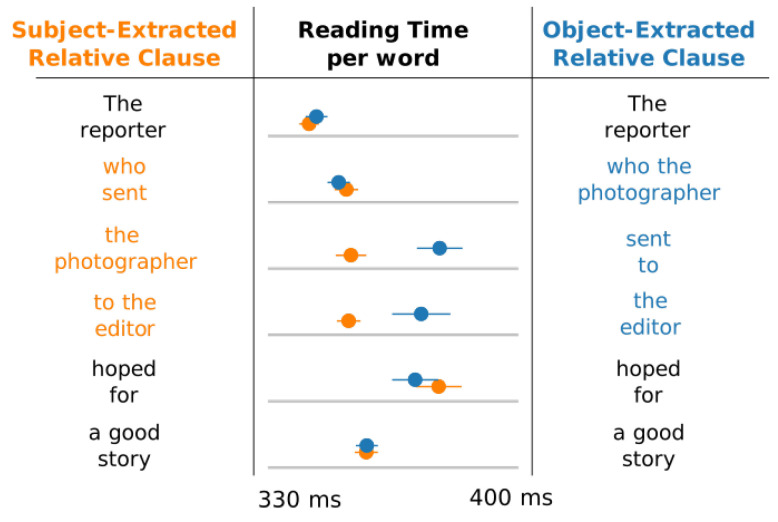
An illustration of reading times in self-paced reading in object-extracted relative (blue) and subject-extracted (orange) relative clauses (adapted from Ref. [3]). While reading times are similar for the main verb (’hoped’), they vary substantially for the embedded one (’sent’). This suggests that the processing load for object-extracted relative clauses diverges most from subject-extracted relative-clauses at the embedded verb.

**Figure 4 entropy-22-00446-f004:**
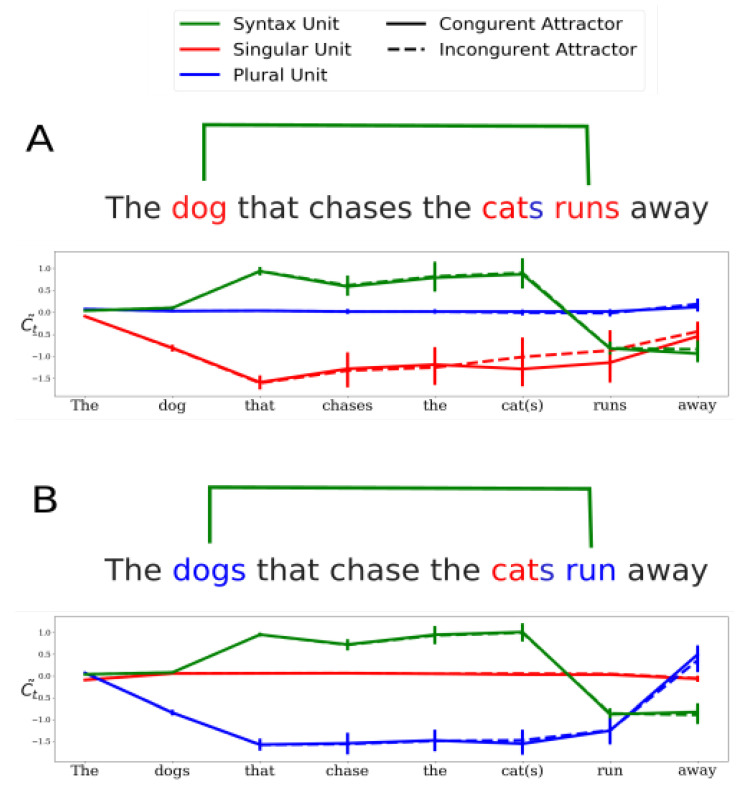
The neural mechanism for long-range grammatical-number agreement. Cell activities of the syntax (green), singular (red) and plural (blue) units are presented during the processing of a sentence with a long-range subject-verb dependency across a subject-extracted relative clause. Panel (**A**) describes unit activity during the processing of sentences in which the main subject is singular. Continuous/dashed lines correspond to cases in which the intervening noun has the same/opposite number as the main subject (i.e., singular/plural). Similarly, Panel (**B**) describes conditions in which the main subject is plural. Error bars represent standard deviation across 2000 sentences.

**Table 1 entropy-22-00446-t001:** Summary of the main elements of the cognitive models for sentence processing. The source for capacity limitation of each theory is described on the right column (see Section 3 for details).

Cognitive Theories for Sentence Processing				
**Theory**	Grammar	Parsing Algorithm	Limiting Resource	Explanation for Capacity Limitation and processing breakdowns
**DLT**	Dependency grammar	Dependency parsing	Energy units	Too many long-range structural integrations take place at a given word, exceeding unit resources.
**ACT-R based**	pCFG	Left-corner	Temporal activity	High similarity among memory items cause unresolvable interference.
**Expectation-based**	pCFG	None	Probability mass	Frequent syntactic structures ‘consume’ most of the probability mass, leading rarer structures to generate high surprisal.
**Symbolic neural architectures**	None	None	Dimensionality	Highly complex syntactic structures require higher state-space dimensionality than that available.
**Neural Language Models**	None	None	Specialized syntax units	The neural circuit for long-range dependencies is sparse and can therefore process only a limited number of nested, or cross, dependencies.

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
