# Peer review of "What Limits Our Capacity to Process Nested Long-Range Dependencies in Sentence Comprehension?"

_entropy, 2020, doi:10.3390/e22040446_

Round 1

Reviewer 1 Report

Broader suggestions:

The authors included a nice review of various previous accounts of the processing difficulty associated with center-embedding and ORC vs. SRCs dating back to Gibson's 1998 model. But I think it would be useful, perhaps in the Discussion, to reflect on even earlier models and their relationship with the authors' NLM account. In the 70s and early 80s, a common intuition was that there was simply not much room in the working memory store, so that if you put a few things on the stack one would have to 'fall off'. For example, Wanner and Maratsos had a well-known ATN model where long-distance dependencies were handled by putting dependents on a HOLD-list of limited capacity. It seems like there are some ways in which the NLM's explanation is similar in spirit, the idea that there is only 'room' to represent one long-distance dependency at a time. But it's also different in that

Second, as the authors mention in their review of the prior literature, one of the criticisms often leveled on classic fixed-capacity syntactic memory explanations was the fact that semantic features of the input seemed to impact capacity, like the fact that you can sometimes get 3 layers of nesting if you use personal pronouns; that was supposed to be one of the big selling points of models that somehow engaged with features of the dependent elements like the ACT-R retrieval-interference models. So it would be useful to go a bit beyond footnote 2 in discussing how the current NLM model will capture that kind of variability.

Third, another case that I think should be discussed somewhere here is center embedding that is not self embedding, like with sentential complement clauses as in 'The idea that the cat that the mouse bit died is ridiculous'. In this kind of case you have a long-distance relation on the outside (idea - is ridiculous) and another one in the middle layer (man - is a spy), but if a bit clunky, it's still way easier to process than the classic cases that only relative clauses. If you could only have one long-distance relation at a time, it seems like the SC/RC embeddings should cause exactly the same problem as the RC/RC ones. On the other hand, Lewis and Vasishth (2005) claimed that the ACT-R based model *can* capture this variation (p. 405-407). So again, since the authors want to make the argument here that the NLM model goes beyond previous models in accounting for the same data with fewer assumptions, I think they should discuss the cases that previous models took as central evidence in their favor.

Finally, I don't want to require the authors to make a major change in their terminology, but I'd like to comment that I really think that 'sparse' is a misnomer here, if it is intended in the way that neuroscientists tend to use it. In neuroscience, 'sparse coding' is usually used to describe the situation where the activity of a relatively small percentage of neurons encode each token in a given domain--but importantly 'small' is not usually intended to be 'one', as 'sparse coding' is supposed to be an improvement upon the most extreme grandmother cell notion. In the current network, the number of 'syntactic' units is actually one, and in their explanation the authors imply that this is critical--if there's only one such unit, you can't accurately encode more than one non-local dependency. But presumably, when we scale this up to the brain, no one is going to want to commit to there truly being only a single neuron--even hard-core grandmother cell theorists like Jeff Bowers don't want to say that.

So in the brain what we will probably have to say is that the constraint is not that there is only one syntactic unit, but that the constraint is that the syntactic units all have to code the same non-local dependency. That is still a useful observation to have emerged from the modeling, I just think it's going to give neuroscientists the wrong idea to make it sound like there's some causal relation between the small number of units and them having to code the same dependency, as the authors do for example in Table 1 when they say that the circuit is sparse and *therefore* can process only a limited number of dependencies, also on p. 10 and p. 15. In the discussion p. 13 it becomes clearer that what the authors mean by sparsity is more like parsimony--not developing the ability to represent two embedded non-local dependencies at a time in a more complex circuit if it's not needed. It may be that this is an alternative meaning of the term sparsity from computer science, but even if so I think if you want a neuroscience audience, it's problematic.

Line-by-line suggestions:

Abstract - I feel like the phrasing 'without a priori postulating a grammar or a syntactic parser' might be misleading--it's not that you know that the neural network didn't develop something equivalent to a grammar/parser, or that it would have developed the same structure if you gave it non-human grammars--it's just that you didn't *specify* the grammar or parser yourself. Could you say instead something like 'without making any a priori assumptions about the form or the grammar or parser'?

p. 1 line 17 - '...combine word sequences into a continuously evolving semantic representation'. Words are not meanings and you're not really combining sequences per se, so this first sentence doesn't really achieve the intended meaning. Maybe instead you could say something more like '...to combine a sequence of word meanings into a continuously evolving...'

p. 1 line 20 - 'the brain must infer whether and how to store and combine a newly presented word' Using the 'infer' and 'inference task' terminology I think could throw many psycholinguistic readers off, as parsing is not the prototypical inference task in that literature. Could you just say 'decide'?

p. 1 line 22 - 'Sentences that contain long-range dependencies among their words are known to be more demanding to process.' Than what? It matters because not all long-range dependencies are equally difficult. Would be safer to qualify with something like '..that contain *certain* long-range dependencies...'

p. 1-2 - I like the simplicity of the exposition of long-distance dependencies, but I wonder if it's a bit misleading. The text at the top of p. 2 implies that the reason for 'reduced reading speed and comprehension' in center-embedding is linear order--the fact that 'the reader must combine the verb with the first noun in the sentence and not its closest neighbor'. But of course it's not just about closest neighbors--a simple sentence with a PP-modified subject like 'The dog next to the cat woke up' also requires combining the verb with the first noun in the sentence and not its closest neighbor, but it is much easier to process. Maybe the simplest fix would be to just leave out the sentence 'When reading the word "ran"...' and let the reader draw their own conclusions about exactly what aspect of non-locality drives the difficulty.

p. 2, line 32 - 'sentence processing dramatically breaks down beyond two levels of nesting'. As noted in example above, processing breaks down not with *any* nesting beyond two levels, but seems like specifically those that involve nesting exactly the same clause type. So maybe safer again to qualify '...beyond two levels of nesting relative clauses'

p. 2, line 36 - 'the same meaning' should be 'a highly related meaning' or something like that. It's not the *same* meaning, sentence (2) asserts something about the dog and sentence (3) does not.

p. 2, line 39 - 'In other words, the syntactic structure directly impacts our cognitive resource demands beyond their associated semantic complexity'. I wouldn't insist that the authors fix this, but exactly because the meanings are not identical, I think this strong claim does go beyond the evidence provided. The incremental semantic complexity of 'the dog that the cat that the mouse...' might well be greater than that of 'the mouse bit the cat that chased the dog...' in the sense that three semantic referents need to be accessible for simultaneous predication in the first (I need to predicate something of the dog, the cat, and the mouse), while it's plausible that only two need to be accessible for predication at any one time in the second (at 'the mouse bit the cat', I update the mouse referent with the predicate {bit cat[x]} and the cat referent with the predicate {bitten by mouse[y]}, and then I can put the mouse[y] referent aside and focus on updating the cat[x] referent only, etc.)

p. 3, line 81 - 'structure of the latent structure' awkward, could just be 'latent structure'

p. 3, line 85 - not to be picky but, a parsing algorithm doesn't technically have to generate structure incrementally!

p. 3, line 89 - says 'resources-based' but line 91 says instead 'memory-based'

p. 4 'These locality effects suggest that longer dependencies are harder to process'. could this be reworded to 'These locality effects motivate the hypothesis that longer dependencies are harder to process'? I worry that 'suggest' is misleading as it makes it sound like the *only* plausible reason for one analysis to be ranked above another in case of ambiguity is their relative processing difficulty, which of course isn't the case.

p. 9, line 178-185 - The limitations of the previous theories described here seemed so weak as to actually hurt the authors' case. I don't see why it is a 'limitation' of a theory that it makes assumptions about the form of the human parsing algorithm, just because there is not yet consensus on that yet. It is well known that different parsing algorithms vary in memory load and processing time for particular sentences, so seems a priori reasonable to think that the explanation for center-embedding difficulty is critically dependent on the parsing algorithm. I also don't see why the existing theories would be unable to provide predictions about neural responses--it will just require adding a linking hypothesis. In my view it would be better to just leave this paragraph out and go straight to describing your own framework. Since none of the existing theories have become consensus yet, I don't think you need to try to come up with specific criticisms of them in order to motivate introducing your own theory--you can just talk about the benefits of your own theory.

p. 13, line 252 - It's not clear to the reader what is meant by 'ORCs are relatively difficult for the network' and yet 'the network successfully performs the task'. I think it would be helpful to say whether the difficulty means relative accuracy or something else.

p. 14, line 318 - 'Brain connectivity analysis is thus predicted to reveal projections from syntactic regions only onto higher-level cortical regions, in which long-range number units are predicted to reside' - I couldn't understand what was intended here. Why do syntactic units need to be in a different region that the long-range number units? And what are these higher-level cortical regions? Is this something about non-linguistic number vs. syntactic number???

p. 15, line 369 - 'resources-based theories...capacity limitation arises due to resource depletion' - I'm not sure resource depletion is the best characterization of retrieval-based theories. ACT-R has a decay mechanism, but not all versions do. The core intuition of most of those theories is more about confusion/errors, or 'noisy retrieval' than it is about resource depletion.

Reviewer 2 Report

This is an interesting paper proposing a possible mechanism, using artificial neural networks, for why center embedded constructions are hard (sentences like “The rat that the cat that the dog chased ran away showed up.”) It follows up on empirical work by Lakretz et al, who fascinatingly show how a neural model can learn long distance number agreement, at a mechanistic level.

There’s a lot of good stuff here, and it’s written clearly and fluidly. In particular, there is an interesting discussion of various psycholinguistic theories (DLT, surprisal, etc) and how they predict different difficulty with these kind of long-distance dependencies.

My concern is that I’m left not knowing entirely what kind of paper this is. It’s submitted as an Opinion, but it actually seems to be a paper with a big empirical claim. But there are no experiments or new data or new analyses presented.

So I think it’s more of a review/position paper. But it’s a review/position paper that makes a big-time empirical claim: that processing difficulty like what’s discussed can be explained by the kind of long-distance and short-distance nodes in the Lakretz et al. paper. Making this claim without evidence seems problematic. 

If true, this claim would be really exciting and have major implications in psycholinguistics. Specifically, there’s a lot of value in the observation that the kind of local neural mechanisms used here could potentially give rise to higher level processing constraints. I hope this continues to be fleshed out.

But I think that a big claim like this needs more evidence before it can be published as a claim. For instance, does the neural network have to be so sparse for long-distance agreement? Or is it like that because these things are rare and so it didn’t bother to learn it? I.e., could you train a model with redundant long-distance agreement nodes and it would then be able to handle these?

I also wasn’t sure about the relationship between what is sometimes called long-distance number agreement nodes and other times just long-distance syntax nodes. There is more to long-distance syntax than just number. The difficulty with sentences like (2) in the paper is not just that it’s hard to get the number agreement right (which is the main difficulty in “key to the cabinet” sentences). It seems to be harder in a general sense. Is the idea that the number nodes are also just long-distance syntax nodes in general. Does that mean that there could never be two long-distance syntactic dependencies active at once?

Also, the authors carefully go through various psycholinguistic theories, but they don’t then test them or directly compare them in an empirical task. The description of the problems with all of these theories is a bit hand-wavy.

Why the maze task? The authors describe one method (a maze task) for getting at processing difficulty. But they don’t run an experiment or show results from a maze task. What is the point of presenting this section?

So, overall: It’s a really exciting area of research. If this were a grant proposal, I’d recommend funding it. But as it is, I think that it should be revised to either be an empirical paper that really drills down into the differential claims of the various theories and shows how a neural approach works better. Or it could be a kind of position paper. But I think the main claim is strong and exciting, but I would urge caution in making it without evidence.

Round 2

Reviewer 2 Report

I think this paper is improved by the changes the authors made. My main concern was not knowing what type of paper this was since it seemed to be making empirical claims, and I think that some small adjustments to the language have made it clear that the point is that, given recent results by this group of authors using neural models, there are intriguing directions to consider for psycholinguistics by taking seriously the idea that these models could actually serve as cognitive models.

It's certainly interesting and potentially important and I'd be happy to see it published. I think we need more cross-talk between deep neural models in computer science and cognitive models, and this is just the kind of thing that advocates for. I hope that, with some more tweaking, it could serve as a primer on various important psycholing models for the deep neural crowd, and it could introduce these kind of models more broadly to the psycholing crowd.

I'd still urge toning down the language in places and making it clear that, whereas other studies show how the other studies by this group show how neural networks encode things like number agreement, this paper does not show that neural networks are good cognitive models. But it does present some evidence as to why they MIGHT be and why it might be worth considering more than they have been considered in psycholinguistics.

I also think that, in places like the new section 4.1.3, there could be more clarity in describing the critical prior work since not everyone will know that work.
